# Expanding syphilis test uptake using rapid dual self-testing for syphilis and HIV among men who have sex with men in China: A multiarm randomized controlled trial

Cheng Wang[1,2☯]*, Jason J. Ong[3,4☯], Peizhen Zhao[1,2☯], Ann Marie Weideman[5,6], Weiming Tang[1,2,7], M. Kumi Smith[8], Michael Marks[3], Hongyun Fu[9], Weibin Cheng[1,2], Fern Terris-Prestholt[10], Heping Zheng[1,2], Joseph D. Tucker[3,7,11]*, Bin Yang[1,2]*

1 Dermatology Hospital of Southern Medical University, Guangzhou, Guangdong, China, 2 Southern Medical University Institute for Global Health and Sexually Transmitted Diseases, Guangzhou, Guangdong, China, 3 Faculty of Infectious and Tropical Diseases, London School of Hygiene & Tropical Medicine, London, United Kingdom, 4 Central Clinical School, Monash University, Victoria, Melbourne, Australia, 5 Department of Biostatistics, University of North Carolina at Chapel Hill, Chapel Hill, North Carolina, United States of America, 6 Center for AIDS Research Biostatistics Core, University of North Carolina at Chapel Hill, Chapel Hill, North Carolina, United States of America, 7 University of North Carolina Project-China, Guangzhou, Guangdong, China, 8 Division of Epidemiology and Community Health, University of Minnesota Twin Cities, Minneapolis, Minnesota, United States of America, 9 Division of Community Health and Research, Eastern Virginia Medical School, Norfolk, Virginia, United States of America, 10 Department of Global Health and Development, London School of Hygiene & Tropical Medicine, London, United Kingdom, 11 Institute for Global Health and Infectious Diseases, School of Medicine, University of North Carolina at Chapel Hill, Chapel Hill, North Carolina, United States of America

☯ These authors contributed equally to this work.
* wangcheng090705@gmail.com (CW); jdtucker@med.unc.edu (JDT); yangbin101@hotmail.com (BY)

**Data Availability Statement:** De-identified data cannot be made open accessed because of the IRB

## Abstract

### Background

Low syphilis testing uptake is a major public health issue among men who have sex with men (MSM) in many low- and middle-income countries. Syphilis self-testing (SST) may complement and extend facility-based testing. We aimed to evaluate the effectiveness and costs of providing SST on increasing syphilis testing uptake among MSM in China.

### Methods and findings

An open-label, parallel 3-arm randomized controlled trial (RCT) was conducted between January 7, 2020 and July 17, 2020. Men who were at least 18 years of age, had condomless anal sex with men in the past year, reported not testing for syphilis in the last 6 months, and had a stable residence with mailing addresses were recruited from 124 cities in 26 Chinese provinces. Using block randomization with blocks of size 12, enrolled participants were randomly assigned (1:1:1) into 3 arms: standard of care arm, standard SST arm, and lottery incentivized SST arm (1 in 10 chance to win US$15 if they had a syphilis test). The primary outcome was the proportion of participants who tested for syphilis during the trial period and confirmed with photo verification and between arm comparisons were estimated with risk

decision to ensure patient confidentiality. Permission can be requested by contacting the Dermatology Hospital of Southern Medical University (190913767@qq.com).

**Funding:** YB received award from National Natural Science Foundation of China (81772240). The funders had no role in study design, data collection and analysis, decision to publish, or preparation of the manuscript.

**Competing interests:** The authors have declared that no competing interests exist.

**Abbreviations:** CFDA, China Food and Drug Administration; CI, confidence interval; CONSORT, Consolidated Standards of Reporting Trials; COVID-19, Coronavirus Disease 2019; FCS, fully conditional specification; IgA, immunoglobulin A; IgG, immunoglobulin G; IgM, immunoglobulin M; MI, multiple imputation; MSM, men who have sex with men; NRI, nonresponder imputation; RCT, randomized controlled trial; RD, risk difference; SST, syphilis self-testing; STI, sexually transmitted infection; TP, *Treponema pallidum*; VCT, voluntary counseling and testing; WHO, World Health Organization.

differences (RDs). Analyses were performed on a modified intention-to-treat basis: Participants were included in the complete case analysis if they had initiated at least 1 follow-up survey. The Syphilis/HIV Duo rapid test kit was used. A total of 451 men were enrolled. In total, 136 (90.7%, 136/150) in the standard of care arm, 142 (94.0%, 142/151) in the standard of SST arm, and 137 (91.3%, 137/150) in the lottery incentivized SST arm were included in the final analysis. The proportion of men who had at least 1 syphilis test during the trial period was 63.4% (95% confidence interval [CI]: 55.5% to 71.3%, $p = 0.001$) in the standard SST arm, 65.7% (95% CI: 57.7% to 73.6%, $p = 0.0002$) in the lottery incentivized SST arm, and 14.7% (95% CI: 8.8% to 20.7%, $p < 0.001$) in the standard of care arm. The estimated RD between the standard SST and standard of care arm was 48.7% (95% CI: 37.8% to 58.4%, $p < 0.001$). The majority (78.5%, 95% CI: 72.7% to 84.4%, $p < 0.001$) of syphilis self-testers reported never testing for syphilis. The cost per person tested was US $26.55 for standard SST, US$28.09 for the lottery incentivized SST, and US$66.19 for the standard of care. No study-related adverse events were reported during the study duration. Limitation was that the impact of the Coronavirus Disease 2019 (COVID-19) restrictions may have accentuated demand for decentralized testing.

## Conclusions

Compared to standard of care, providing SST significantly increased the proportion of MSM testing for syphilis in China and was cheaper (per person tested).

## Trial registration

Chinese Clinical Trial Registry: ChiCTR1900022409.

Author summary

### Why was this study done?

- Men who have sex with men (MSM) have a high burden of syphilis, while testing coverage remains low.

- Syphilis self-testing (SST) could be useful to help expand syphilis testing among MSM.

- This study is the first randomized controlled trial (RCT) to evaluate the effectiveness and cost of SST.

### What did the researchers do and find?

- We recruited 451 MSM participants in a 3-arm RCT from 124 cities in 26 Chinese provinces between January 7, 2020 and July 17, 2020.

- Our study showed that promoting SST among MSM substantially increased syphilis test uptake compared with the standard of care.

- The cost per person tested is cheaper for the SST arm compared to the standard of care arm.

### What do these findings mean?

- Our RCT and economic evaluation strengthens the evidence for SST programs among MSM.

- The interpretation of our study's findings might be affected by the Coronavirus Disease 2019 (COVID-19) restrictions, which may have accentuated demand for decentralized testing.

- Future studies are needed to enhance linkage to clinical and public health services after an individual uses a self-test kit.

## Introduction

There were an estimated 6.3 million new cases of syphilis globally in 2016 [1]. Countries have noted syphilis increasing particularly among men who have sex with men (MSM) [2]. This higher risk may be related to structural factors such as denser sexual networks and stigma leading to poorer access to health services and individual factors such as condomless sex with multiple partners [3,4]. In China, syphilis incidence has increased from 1.0 to 32.2 per 100,000 between 1995 and 2016 [5]. Regular syphilis testing is a key strategy for syphilis control [6,7]. Earlier diagnosis and treatment prevents morbidity, mortality, and onward transmission [7]. In most countries, syphilis testing occurs in health facilities [8], but existing facility-based syphilis testing and management resources and services are inadequate to curb the spread of syphilis. Studies suggest that only 30% of MSM in China have ever received a syphilis test [9]. Anticipated stigma associated with syphilis, lack of sexual behavior disclosure to health professionals [8], and the Coronavirus Disease 2019 (COVID-19) restrictions make facility testing more difficult. Syphilis self-testing (SST) may be an effective method to address these barriers by complementing and extending facility-based testing [9].

There are efforts to decentralize syphilis testing, including venue-based testing, self-sampling (sending a specimen to a laboratory), and self-testing [9,10]. SST is a process whereby an individual collects their own specimen, performs the test, and interprets the result themselves [4]. An immunochromatographic test uses blood to detect treponemal antibodies using a rapid test [11], similar to blood-based HIV self-testing. Evidence from a large body of blood-based HIV self-testing programs worldwide demonstrates the feasibility, acceptability, and efficacy of decentralized sexually transmitted infections (STIs) testing and increasing testing uptake among MSM [12]. A cross-sectional study of 699 MSM from 21 provinces in China found 48% of MSM who had tested for syphilis used self-testing [4]. This study also reported that 52% of MSM reported that SST was their first syphilis test. Thus, decentralizing syphilis testing by providing more options for home-testing can open up new possibilities to deliver syphilis testing to those in greatest need. However, the policy context for SST and HIV self-testing is different. Although 59 countries have policies supporting the use of HIV self-testing among key populations [13], none have guidelines supporting SST.

In recent years, there is growing interest in using social innovation methods to solve complex problems [14]. For example, crowdsourcing (where a group of individuals solve a problem and solutions are shared with the public) [15] could be combined with the insight that financial incentives could enhance healthy behaviors [16]. There is evidence that financial incentives may improve uptake of HIV/STI testing [17]. We previously reported that MSM living in China—particularly those at higher risk for syphilis—reported that they were more likely to test for syphilis if a lottery-based incentive was available [18]. A lottery-based incentive is a form of financial incentive whereby an individual who receives a syphilis test is enrolled in a chance to win a monetary reward. We conducted a crowdsourcing call with MSM in China to design a lottery-based incentive to examine if the addition of this would further enhance the appeal of SST. The combination of the implementation of SST with lottery-based financial incentives has not been explored.

This study aimed to evaluate the effectiveness and cost of providing SST on increasing syphilis testing uptake among MSM in China compared with standard of care. The primary null hypothesis was no difference between standard SST and the standard of care among MSM in China. Our secondary objective was to examine the difference between lottery incentivized SST and standard SST. Before implementing this trial, we assessed the acceptability, benefits, and harms associated with SST [4], examined participants' ability to follow test instructions and interpret results [19], and evaluated the study design, recruitment process, and materials [19].

## Methods

### Study design and participants

The full study protocol has been published elsewhere (S1 Study Protocol) [19]. This is an open-label, parallel 3-arm randomized controlled trial (RCT) with individuals randomized in a 1:1:1 ratio to 3 study arms: control arm (standard of care); standard SST arm; and lottery incentivized SST arm. Control arm participants received information on self-referral pathways for free facility-based syphilis testing. Both self-testing arms were offered access to dual syphilis/HIV self-test kits for free at monthly intervals via mail.

Recruitment took place from January 7, 2020 to January 17, 2020. Participants in each arm were followed every 3 months for 6 months. The trial follow-up and data collection were completed on July 17, 2020. The follow-up was conducted during the COVID-19 pandemic when facility-based syphilis testing was still available with a series of risk assessment procedures before patients could enter health facilities.

The study was approved by the ethics review committee of the Southern Medical University Dermatology Hospital (GDDHLS-20181206). We reported our studies according to the Consolidated Standards of Reporting Trials (CONSORT) guidelines (S1 CONSORT Checklist), and the CONSERVE 2021 statement (S1 CONSERVE Checklists). The trial was registered with the Chinese Clinical Trial Registry, number ChiCTR1900022409.

### Syphilis self-testing kit

The practice of HIV self-testing among MSM living in China is acceptable, feasible, and safe [20,21]. SST kits can be accessed on e-commerce platforms or through existing HIV self-testing programs in China [4]. There are 10 brands of SST kits available on the 2 largest e-commerce platforms in China. All the kits use blood to detect treponemal-specific antibodies with a colloidal gold method, which cannot distinguish between current and past infection. In this trial, we used the syphilis and HIV combo rapid test kit because of its excellent test characteristics (sensitivity of 99.7% and specificity of 99.7%) and the World Health Organization (WHO)

prequalification [22]. The Syphilis/HIV Duo test is a solid phase immunochromatographic assay for the qualitative detection of antibodies to all isotypes (immunoglobulin G [IgG], immunoglobulin M [IgM], and immunoglobulin A [IgA]) specific to HIV-1/2 and/or *Treponema pallidum* (TP) simultaneously in human serum, plasma, or whole blood. The procedure of blood-based SST is similar to the procedure for blood-based HIV self-testing. All the kits were certified by the China Food and Drug Administration (CFDA) with excellent sensitivity and specificity [4]. The cost per test kit ranges from US$2.5 to US$15.

## Procedures

Fig 1 shows the key concepts of the interventions for each study arm.

**Recruitment.** We recruited participants from a large MSM-oriented mobile social app—Blued (Danlan, Beijing, China) [23]. Blued is China's most popular social networking mobile application among MSM. By February 2018, Blued had 40 million registered users globally, with 70% in Mainland China [24]. An invitation to join the study was posted on Blued's portal website and startup screen and was also sent via other social media portals (WeChat, a popular Chinese messaging app; Weibo, a microblogging platform; and QQ, a messaging platform). Individuals who clicked the advertisement were directed to a survey website hosted by Sojump (Changsha Haoxing Information Technology, China) to give consent and complete eligibility screening. Participants were eligible if they were born biologically male, aged 18 years or older, had condomless anal sex with other men in the past 12 months, reported not testing for syphilis in the last 6 months, planned to live in China for the next 6 months, and had a stable residence where they could securely receive a postal package. Participants were excluded if they participated in another research program related to HIV/STIs during the study period or could not provide consent.

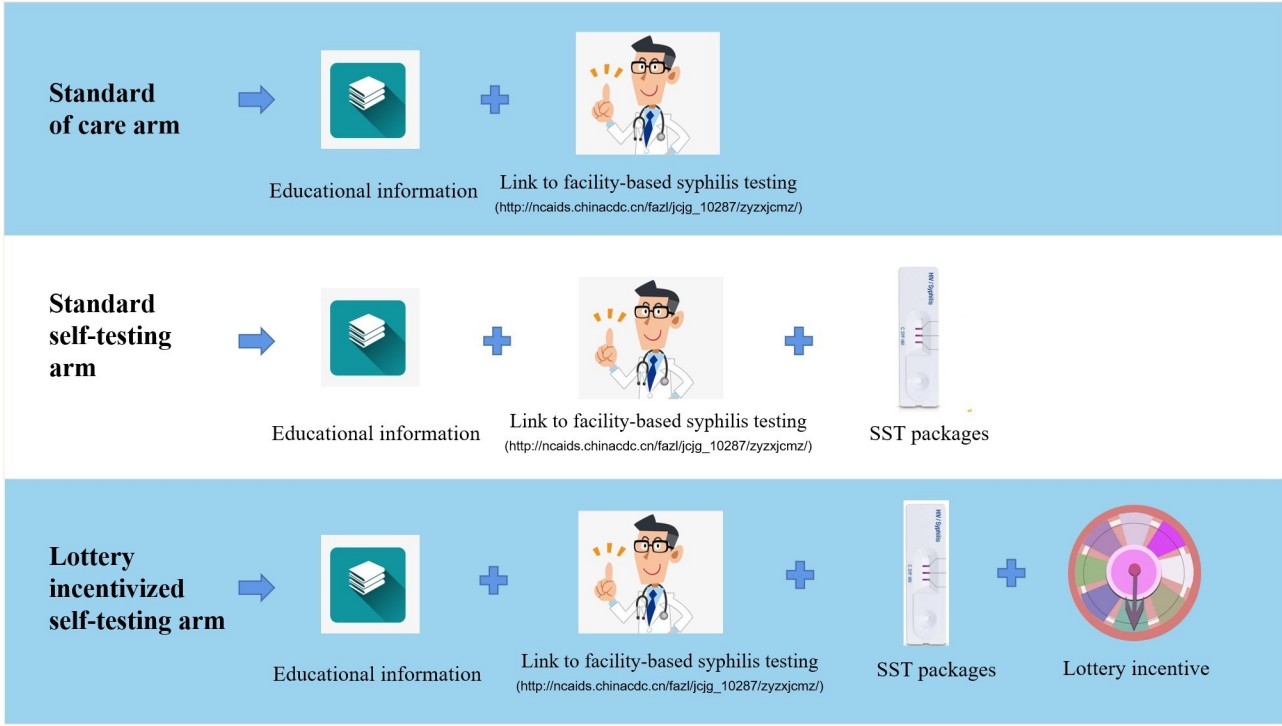

**Fig 1. Key concepts of the 3 study arms: standard of care, SST, and lottery incentivized self-testing.** SST, syphilis self-testing.

Eligible participants completed a baseline survey and were required to provide contact information including a cell phone number or a WeChat account to receive text messages, a shipping address for participants who applied for kits, and a preferred name, which were all stored in a password-protected file that a designated research assistant could only access. All participants provided online informed consent before the baseline survey by clicking on a "button" indicating that the participant has read the consent form and agrees to participate. We enrolled participants from 124 cities in 26 Chinese provinces (S1 Table). Participants were provided with US$3 for their time completing the baseline survey.

**Randomization and allocation.** Men who completed the baseline survey and provided contact information were directed to contact a research assistant responsible for allocating participants into one of the 3 study arms. Participants were randomly assigned (1:1:1) to one of the 3 study arms using block randomization with a block size of 12. Computer-generated randomization codes were produced by a biostatistician using SAS version 9.4 (SAS Institute, Cary, North Carolina, United States of America) who was not involved in participant enrollment. An independent research assistant who was not involved in the trial sealed the study allocation in numbered opaque envelopes. The study design required participants and recruitment staff to be aware of the study arm allocation; however, they did not know the arm assigned until an envelope was opened after informed consent and enrollment.

**Intervention.** Men in the standard of care arm were provided with an educational message including the risk of acquiring syphilis, the importance of screening for syphilis, a link to resources to locate local STIs or HIV voluntary counseling and testing (VCT) clinics, and encouragement to screen for syphilis testing at their local VCT clinics every 3 months. VCT clinics in China provide both HIV and syphilis testing free of charge. The educational message was sent by SMS or WeChat quarterly: at enrollment, 3 months, and 6 months.

In addition to the message for men in the standard of care arm, men in the standard SST arm were also provided with SST services whereby men could order a maximum of one SST package for free per month through WeChat during the study period. The SST package was sent to men by postal mail (express delivery), which arrived at the intended destination within an average of 2.5 days. The package included the manufacturer-supplied step-by-step instructions, a web link to an instructional video, and a result report card (S1 Text, S1, S2 and S3 Figs). The result report cards allowed participants to upload a photo of their test results to a verification platform by scanning a barcode on the card that was uniquely linked to the study participant.

Men in the lottery incentivized SST arm were provided with the standard SST arm intervention. In addition, men who conducted the SST or facility-based testing during the trial and uploaded photo verification of the results were offered an opportunity to participate in a lottery draw. The lottery draw provided a 10% chance of winning US$15 in cash. Each man could participate in the lottery draw up to 2 times per month during the trial if they uploaded both self-testing and facility-based testing results. An open crowdsourcing contest determined the procedure of the lottery draw and the images that would be shown to men in the lottery incentivized SST arm. The crowdsourcing contest resulted in 14 lottery schemes from which the winning entries were then used to develop the lottery draw program in this trial. A total of 20 volunteer MSM voted on the amount of prize money and the probability of winning in the lottery draw.

**Data collection.** In all 3 study arms, men were encouraged to upload syphilis testing record and results. Confirmed SST was determined by photo verification of the used test kit. Facility-based syphilis testing and results were determined by photo verification of the test report. A research assistant conducted the verification process by checking the photos uploaded in the verification platform or WeChat. Men in all 3 arms who uploaded photo verification were provided with US$1.

All participants completed a baseline questionnaire at enrollment, which included sociodemographic characteristics, sexual behaviors, past HIV/syphilis testing and treatment, and attitude toward SST. During follow-up, we invited all participants to complete a brief online questionnaire at 3 and 6 months that collected information on the number, time, location, reasons for, and results of syphilis tests (including self-tests) and other STI tests at clinics and community sites, and sexual risk behaviors in the past 3 months. Participants in the 2 self-testing arms were also asked to report the number of self-tests that were used for personal testing, to test a partner, or given to someone else. Participants were provided with US$5 for their time for completing each follow-up survey.

**Follow-up support.** A syphilis counselor was available for support through WeChat and telephone from 8:00 AM to 5:30 PM, Monday through Friday. Support included giving pretest counseling, instructing how to use the self-test kit, helping to interpret results, and providing advice for reactive test results. Participants were advised to inform the counselor immediately if they had a reactive self-test result. Any participant with a reactive self-test result was referred for confirmatory laboratory testing and clinical examination at the designated clinics or hospitals. A research assistant undertook further follow-up to obtain confirmatory testing results and treatment information for men diagnosed with syphilis. A log of each enquiry was recorded.

## Outcomes

The primary outcome was the proportion of participants tested for syphilis either at a clinic or via self-test during the whole 6-month study period, excluding confirmatory tests after a reactive self-test between the intervention groups and the control group. This was assessed in all participants using photo verification. Secondary outcomes included syphilis testing frequency during the trial, number of newly identified syphilis infections during the trial, the linkage to syphilis clinical care after self-testing during the trial, the proportion of HIV and other STIs (chlamydia, gonorrhea, human papillomavirus, and herpes simplex virus) tests during the trial, risky sexual behaviors in the last 3 months (such as number of sex partners, anal group sex, substance use before or during sex, and condomless anal sex). The secondary outcomes were assessed based on self-reported data from each follow-up survey. We also reported the total and incremental unit costs for each arm. The costs collected were from a healthcare provider perspective. Details of the economic evaluation are summarized in S1 Appendix (p. 11–20).

## Statistical analysis

**General analysis.** We used descriptive analyses to report the demographic and behavioral characteristics of participants in each study arm. All inferential tests were 2-sided with a type 1 error level of 0.05. Data analysis was conducted as modified intention to treat: Participants were included in the complete case analysis if they had initiated at least 1 follow-up survey. Risk differences (RDs) and corresponding 95% confidence intervals (CIs) were expressed as percentages. All analyses were conducted using SAS version 9.4 (SAS Institute).

**Analysis of primary outcome.** The primary outcome of test uptake was evaluated as a difference in the probability of syphilis testing during the trial between the 2 groups. For binomial proportions presented in the abstract, since the sample size was sufficient and there were no extreme proportions (close to 0 or 1), $p$-values and 95% CIs were obtained via a Wald test. However, due to small or zero cell counts and proportions, RDs were presented with $p$-values generated from Barnard unconditional exact test and 95% CIs from the Chan and Zhang method (which is based on a score statistic) [25]. We also conducted sensitivity analyses on

uptake of syphilis testing using nonresponder imputation (NRI) to impute the worst- and best-case scenarios and multiple imputation (MI) by fully conditional specification (FCS) with logistic regression. For NRI, the sensitivity analyses involved replacing missing values with the worst value (not tested for syphilis) and best value (tested for syphilis) in the observed data. We assumed for the best value that all the men with missing follow-up data who applied for the SST kits received syphilis testing during the trial. For MI, Rubin's rule [26] was used to compute pooled estimates for the RDs and standard errors. The number of imputations was selected based on relative efficiency such that $FMI_{max}/m \approx 0.01$, where $FMI_{max}$ is the maximum fraction of missing information (computed as in [27]) and $m$ is the number of imputations. The FMI ranged from 0.02 to 0.21, so $m = 20$ was chosen for the number of imputations. Compared to those who completed at least 1 follow-up survey, participants lost-to-follow-up differed on 3 demographic variables included in the imputation model (S2 Table): annual income, disclosure as MSM to family, friends, or healthcare professionals, and ever tested for HIV.

**Analysis of secondary outcome.** The secondary outcome of male sex partners in the past 3 months was reported as the mean number of partners in each arm. For the mean differences between groups, *p*-values and 95% CIs were computed using a Satterthwaite approximation (assuming unequal variances between the groups). Other secondary outcomes, including the proportion of HIV and other STIs and risky sexual behaviors in the last 3 months, were evaluated using the same methods described for the primary outcome.

**Economic evaluation.** We used a microcosting approach to estimate the financial cost from a health provider perspective. Costs were collected alongside the trial and categorized as either fixed or variable. For fixed costs (i.e., independent of the number of tests conducted), we estimated the cost of capital (building rent), personnel support, and office equipment (Table A in S1 Appendix). For the lottery incentivized self-testing arm, we included the cost of developing the lottery strategy from a crowdsourcing activity—this was annualized over an expected useful life of 5 years, using a discount rate of 3%. For variable costs (i.e., dependent on the number of tests conducted), we estimated the cost of supplies used for syphilis testing. All costs were reported in 2020 USD based on the exchange rates using OANDA currency conversions (1 USD = 6.96 yuan). We analyzed the cost in Microsoft Excel 2019 (Microsoft, USA). Using parameters from Table A in S1 Appendix, which were informed by the trial, we created a decision tree model using TreeAge Pro 2020 (TreeAge Software, Inc., Williamstown, MA) to explore the cost-effectiveness of the 3 arms (Fig A in S1 Appendix).

## Results

Overall, the survey link was clicked 54,082 times; 52,089 withdrew before reading the consent form, and 2,713 were screened for study eligibility (Fig 2). In total, 2,124 men were ineligible because they reported no condomless anal sex with men in the last year (*n* = 1,362), had tested for syphilis in the last 6 months (*n* = 600), were younger than 18 years old (*n* = 82), currently had syphilis (*n* = 55), or were not born biologically male (*n* = 25). An additional 138 men were excluded for not signing the consent form (*n* = 38) or not providing contact information (*n* = 100).

A total of 451 men were enrolled and randomly assigned to one of the study arms: 150 men in the standard of care arm, 151 men in the standard SST arm, and 150 men in the lottery incentivized SST arm. During the trial, 111 (73.5%) men in the standard SST arm and 117 (78.0%) men in the lottery incentivized SST arm applied for the SST kits at least once. All the applicants received the mailed SST kits. Eight men asked for help on the use of SST or interpretation of the outcome of SST during the trial. During the study, 136 (90.7%) in the standard of

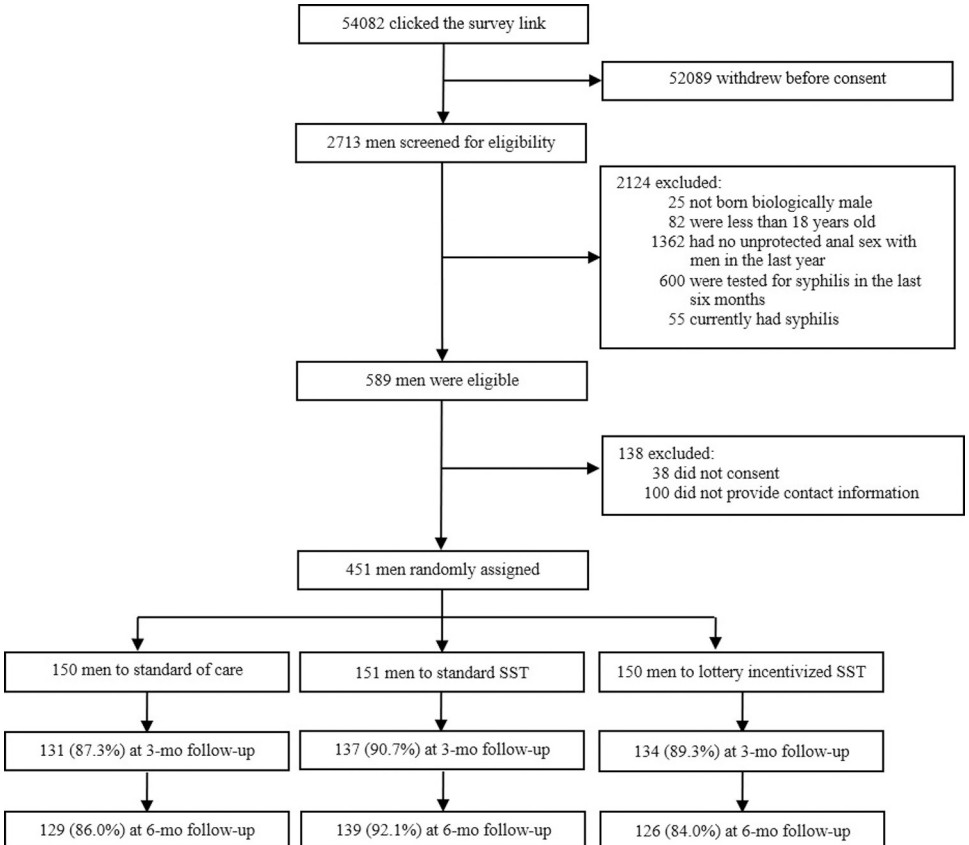

**Fig 2. CONSORT flowchart. CONSORT, Consolidated Standards of Reporting Trials; SST, syphilis self-testing.**

care arm, 142 (94.0%) in the standard of SST arm, and 137 (91.3%) in the lottery incentivized SST arm initiated at least 1 follow-up survey and were included in the final analysis (Fig 2). Among them, 48.2% (200/415) of participants uploaded a photo of their test results, and 93.5% (187/200) of those photos matched the self-reported results. Characteristics of participants lost to follow-up (36 men) differed in disclosure of sexual orientation and annual income from those who completed at least 1 follow-up survey (415 men) during the trial period (S2 Table).

Demographic characteristics were similar across the 3 arms (Table 1). Most participants were 30 years of age or younger (79.4%, 358/451), never married (87.1%, 393/451), had at least a college degree (70.3%, 317/451), and had an annual income of US$9,500 or below (69.3%, 312/451). Around half (50.3%, 227/451) reported ever disclosing their sexual orientation. About two-thirds of men (69.2%, 312/451) reported that they had multiple sexual partners in the past 3 months, and 376 (83.4%) men reported having had condomless anal sex in the past 3 months. In total, 309 (68.5%) men had ever tested for HIV, while 91 (20.2%) had ever tested for syphilis. Only 36 (8%) men had ever used syphilis self-test (Table 1).

Overall, 90 (63.4%) of 142 men in the standard SST arm underwent at least 1 syphilis test during the trial period compared to 90 (65.7%) of 137 men in the lottery incentivized SST arm and 20 (14.7%) of 136 men in the standard of care arm (Table 2). The proportion of individuals who underwent syphilis testing during the trial period was higher in the standard SST arm compared to standard of care (RD: 48.7%, 95% CI: 37.8% to 58.4%, $p < 0.001$) and in the lottery incentivized SST arm compared to standard of care (RD: 51.0%, 95% CI: 40.2% to 60.6%, $p < 0.001$). Compared to the standard SST arm, there was no observed difference in the

**Table 1. Baseline social demographic and behavioral characteristics of MSM in China.**

| | Total (*N* = 451) | SOC (*N* = 150) | Standard SST (*N* = 151) | Lottery incentivized SST (*N* = 150) |
|---|---|---|---|---|
| **Age (years)** | | | | |
| ≤30 | 358/451 (79.4) | 115/150 (76.7) | 126/151 (83.4) | 117/150 (78.0) |
| >30 | 93/451 (20.6) | 35/150 (23.3) | 25/151 (16.6) | 33/150 (22.0) |
| Mean (SD) | 25.6 (6.3) | 25.6 (6.5) | 25.6 (6.4) | 25.7 (6.2) |
| **Marital status** | | | | |
| Never married | 393/451 (87.1) | 128/150 (85.3) | 134/151 (88.7) | 131/150 (87.3) |
| Ever married | 58/451 (12.9) | 22/150 (14.7) | 17/151 (11.3) | 19/150 (12.7) |
| **Annual income (US$)** | | | | |
| <3,000 | 85/451 (18.9) | 32/150 (21.3) | 23/151 (15.2) | 30/150 (20.0) |
| 3,000 to 6,000 | 85/451 (18.9) | 25/150 (16.7) | 31/151 (20.5) | 29/150 (19.3) |
| 6,001 to 9,500 | 142/451 (31.5) | 47/150 (31.3) | 47/151 (31.1) | 48/150 (32.0) |
| 9,501 to 12,500 | 78/451 (17.3) | 26/150 (17.3) | 28/151 (18.5) | 24/150 (16.0) |
| ≥12,501 | 61/451 (13.5) | 20/150 (13.3) | 22/151 (14.6) | 19/150 (12.7) |
| **Highest education** | | | | |
| High school or below | 134/451 (29.7) | 39/150 (26.0) | 48/151 (31.8) | 47/150 (31.3) |
| College or beyond | 317/451 (70.3) | 111/150 (74.0) | 103/151 (68.2) | 103/150 (68.7) |
| **Disclosure as MSM to family, friends, or healthcare professionals** | | | | |
| Never | 224/451 (49.7) | 84/150 (56.0) | 64/151 (42.4) | 76/150 (50.7) |
| Ever | 227/451 (50.3) | 66/150 (44.0) | 87/151 (57.6) | 74/150 (49.3) |
| **Number of male sex partners in the past 3 months** | | | | |
| 0 to 1 | 139/451 (30.8) | 40/150 (26.7) | 48/151 (31.8) | 51/150 (34.0) |
| Multiple | 312/451 (69.2) | 110/150 (73.3) | 103/151 (68.2) | 99/150 (66.0) |
| Mean (SD) | 2.8 (2.3) | 2.9 (2.3) | 2.8 (2.4) | 2.5 (2.0) |
| **Anal sex without the use of condom in the past 3 months** | | | | |
| No | 75/451 (16.6) | 27/150 (18.0) | 27/151 (17.9) | 21/150 (14.0) |
| Yes | 376/451 (83.4) | 123/150 (82.0) | 124/151 (82.1) | 129/150 (86.0) |
| **Ever tested for HIV** | | | | |
| No | 142/451 (31.5) | 55/150 (36.7) | 37/151 (24.5) | 50/150 (33.3) |
| Yes | 309/451 (68.5) | 95/150 (63.3) | 114/151 (75.5) | 100/150 (66.7) |
| **Ever tested for syphilis** | | | | |
| No | 360/451 (79.8) | 124/150 (82.7) | 117/151 (77.5) | 119/150 (79.3) |
| Yes | 91/451 (20.2) | 26/150 (17.3) | 34/151 (22.5) | 31/150 (20.7) |
| **Ever used a syphilis self-test** | | | | |
| No | 415/451 (92.0) | 141/150 (94.0) | 139/151 (92.1) | 135/150 (90.0) |
| Yes | 36/451 (8.0) | 9/150 (6.0) | 12/151 (7.9) | 15/150 (10.0) |

Data are *n*/*N* (%) unless otherwise indicated.

MSM, men who have sex with men; SOC, standard of care; SST, syphilis self-testing.

proportion of individuals testing for syphilis in the lottery incentivized SST arm during the first 3 months (RD: −0.3%, 95% CI: −12.2% to 11.8%, *p* = 1.00). However, there was an observed difference between the standard SST and lottery incentivized SST arm during the second 3 months of follow up (RD: 18.1%, 95% CI: 3.8% to 29.9%, *p* = 0.003) (Table 2). Sensitivity analyses for the primary outcome returned similar results (S3 Table).

The proportion of men who tested for syphilis more than once during the trial was higher in the standard SST arm compared to standard of care (RD: 25.2%, 95% CI: 17.2% to 33.7%, *p* < 0.001) and in the lottery incentivized SST arm compared to standard of care (RD: 40.1%,

**Table 2. Uptake of syphilis testing among participants who initiated at least 1 follow-up survey.**

| | SOC | Standard SST | Lottery incentivized SST | Standard SST versus SOC | | Lottery incentivized SST versus SOC | | Lottery incentivized SST versus standard SST | |
|---|---|---|---|---|---|---|---|---|---|
| | *n/N* (%) | *n/N* (%) | *n/N* (%) | RD (95% CI) | *p*-Value | RD (95% CI) | *p*-Value | RD (95% CI) | *p*-Value |
| **Proportion of participants who tested for syphilis (primary outcome)** | | | | | | | | | |
| Month 0 to 3 | 7/131 (5.3) | 74/137 (54.0) | 72/134 (53.7) | 48.7 (38.9, 57.8) | <0.0001 | 48.4 (38.3, 57.6) | <0.0001 | −0.3 (−12.2, 11.8) | 1.00 |
| Month 4 to 6 | 14/129 (10.9) | 51/139 (36.7) | 69/126 (54.8) | 25.8 (15.6, 35.7) | <0.0001 | 43.9 (33.2, 54.0) | <0.0001 | 18.1 (3.8, 29.9) | 0.003 |
| Overall | 20/136 (14.7) | 90/142 (63.4) | 90/137 (65.7) | 48.7 (37.8, 58.4) | <0.0001 | 51.0 (40.2, 60.6) | <0.0001 | 2.3 (−9.0, 13.6) | 0.77 |
| **Participants by the total of syphilis testing during the trial (secondary outcome)** | | | | | | | | | |
| 0 | 116/136 (85.3) | 52/142 (36.6) | 47/137 (34.3) | −48.7 (−58.4, −37.8) | <0.0001 | −51.0 (−60.6, −40.2) | <0.0001 | −2.3 (−13.6, 9.0) | 0.77 |
| 1 | 16/136 (11.8) | 50/142 (35.2) | 31/137 (22.6) | 23.5 (12.9, 33.2) | <0.0001 | 10.9 (1.75, 20.0) | 0.02 | −12.6 (−23.1, −1.6) | 0.02 |
| 2 | 3/136 (2.2) | 25/142 (17.6) | 29/137 (21.2) | 15.4 (8.6, 22.9) | <0.0001 | 19.0 (11.6, 26.9) | <0.0001 | 3.6 (−5.9. 13.2) | 0.53 |
| 3 | 1/136 (0.7) | 8/142 (5.6) | 20/137 (14.6) | 4.9 (0.7, 10.2) | 0.02 | 13.9 (8.0, 21.0) | <0.0001 | 9.0 (1.7, 16.8) | 0.01 |
| 4 | 0/136 (0.0) | 6/142 (4.2) | 8/137 (5.8) | 4.2 (1.1, 9.0) | 0.02 | 5.8 (2.3, 11.2) | 0.004 | 1.6 (−4.0, 7.6) | 0.56 |
| >4 | 0/136 (0.0) | 1/142 (0.7) | 2/137 (1.5) | 0.7 (−2.2, 3.9) | 0.52 | 1.5 (−1.4, 5.3) | 0.21 | 0.8 (−2.6, 4.7) | 0.60 |
| **Participants who tested more than 1 time during the trial (secondary outcome)** | | | | | | | | | |
| >1 | 4/136 (3.0) | 40/142 (28.2) | 59/137 (43.1) | 25.2 (17.2, 33.7) | <0.0001 | 40.1 (31.2, 49.2) | <0.0001 | 14.9 (2.4, 26.0) | 0.01 |

RD = risk (probability) difference expressed as a percentage.

CI, confidence interval; RD, risk difference; SOC, standard of care; SST, syphilis self-testing.

95% CI:% 31.2 to 49.2%, *p* < 0.001). Additionally, this proportion was higher in the lottery incentivized SST arm than standard SST (RD: 14.9%, 95% CI: 2.4% to 26.0%, *p* = 0.01) (Table 2).

Table 3 summarized the economic evaluation: The cost per person tested was US$66.19 for standard of care, US$26.55 for SST, and US$28.09 for lottery incentivized SST. The incremental cost per person tested for SST compared to standard of care was US$17.55 per person tested and for lottery incentivized SST compared to SST was US$35.55 per person tested. Fig B in S1 Appendix demonstrated that the incremental cost per person tested comparing SST with standard of care would decrease the most with lower (variable and fixed) costs for SST and higher fixed costs of standard of care. Fig C in S1 Appendix showed that the standard of care option was most likely to be cost-effective when the willingness to pay was less than US$17 per case tested, and SST was most likely to be cost-effective between a willingness to pay of US$17 and

**Table 3. Incremental cost per additional person tested and per additional case detected of syphilis testing strategies among MSM in China.**

| | Unit cost | Effectiveness | Cost-effectiveness ratio |
|---|---|---|---|
| **Proportion of men tested** | | | |
| SOC | US$5.60 | 0.0846 | |
| SST | US$12.12 | 0.4565 | US$17.55 per additional man tested |
| Lottery | US$15.45 | 0.5500 | US$35.55 per additional man tested |
| **Case positivity** | | | |
| SOC | US$5.60 | 0.0039 | |
| SST | US$12.12 | 0.0254 | US$303.27 per additional case detected |
| Lottery | US$15.45 | 0.0308 | US$614.56 per additional case detected |

Lottery, lottery incentivized syphilis self-testing; MSM, men who have sex with men; SOC, standard of care; SST, syphilis self-testing.

US$36 per case tested. These findings highlight the benefits of SST from a cost-effectiveness standpoint. The complete economic evaluation was provided in the S1 Appendix.

Of the 200 participants tested for syphilis, 191 (95.5%) men were SST, and 150 (78.5%) men reported that the self-test was their first ever syphilis test. A total of 14 (7.0%, 14/200) men were found to be infected with syphilis, 5 of whom (35.7%, 5/14) were newly identified through this study (1 was in the standard of care arm, 2 in the standard SST arm, and 2 in the lottery incentivized arm), and all 5 reported receiving further confirmatory testing and treatment (S4 Table). Additionally, among those who self-tested for syphilis, 18 (9.4%, 18/191) men were found to be infected with HIV, and 8 men (44.4%, 8/18) were newly identified through this study. The proportion of men who conducted syphilis testing by different testing mode (self-testing versus facility-based testing) are summarized in S5 Table.

Among the men who had syphilis testing during the trial, the proportion of individuals testing for other STIs during the trial in the standard SST arm was significantly lower than in the standard of care arm (RD: −24.4%, 95% CI: −48.5% to −0.9%, $p$ = 0.02) (Table 4).

## Discussion

Our RCT found that promoting SST among MSM substantially increased syphilis test uptake compared with the standard of care. We also explored any further benefits from adding a financial incentive, but we observed that the additional benefits from lottery incentivized SST were marginal. To our knowledge, this study is the first RCT to evaluate the effectiveness and cost of SST and extends the limited literature on SST among MSM [4,19,28]. SST could complement other efforts to decentralize syphilis testing such as syphilis self-collection and venue-based testing [9].

Our study showed that the SST strategy could increase syphilis testing uptake among MSM. This finding is consistent with the results of our pilot RCT [19] and a number of studies on HIV self-testing among MSM [3,29]. This effect of SST might be related to the increasing availability of online SST kits, and the growing acceptability of self-testing due to widespread public health and community-based organization programs to promote HIV self-testing. Additionally, we found that the majority (78.5%) of syphilis self-testers reported that the self-test was their first ever syphilis test. This suggests that SST could help increase first-time testing among individuals who do not seek testing in a facility-based setting. Better understanding first-time testers within sexual networks is critical for the control of syphilis [30]. Our study observed that those tested for syphilis/HIV in the intervention arms were less likely to be tested for other STIs. However, the benefits of improving the uptake of HIV and syphilis testing might outweigh the cost of not screening for other STIs in asymptomatic persons [31]. Further research is warranted to evaluate the impact of a likely reduction in testing for other STIs if HIV/SST is scaled up.

We found that SST is cheaper per person tested compared to facility-based testing. While numerous economic research exists for HIV-self testing [32,33], little is known about the costs associated with implementing SST. We only found one current estimate of the unit cost for syphilis management in China of US$124 (USD, 2011) [34]. Therefore, our economic evaluation data are important for decision-makers to efficiently and fairly allocate limited resources. We likely underestimated the value of screening because we did not include the benefit of averting ongoing syphilis transmissions. Another study has highlighted the value of frequent syphilis testing reducing onward transmission of syphilis [7]. Our study found a higher proportion of men in the intervention arms who tested for syphilis more than once during the trial period. Together, this further reinforces the economic value of investing in syphilis screening programs that encourages regular syphilis testing among those at risk.

**Table 4. HIV/STI testing and sexual behaviors self-reported by men who had syphilis testing during the trial and initiated at least 1 follow-up survey.**

| SOC group | Standard SST group | Lottery incentivized SST group | Standard SST versus SOC | | Lottery incentivized SST versus SOC | | Lottery incentivized SST versus standard SST | |
|---|---|---|---|---|---|---|---|---|
| Mean (SD) | Mean (SD) | Mean (SD) | MD (95% CI)[a] | p-Value | MD (95% CI) | p-Value | MD (95% CI) | p-Value |
| **Number of male sex partners in the past 3 months** | | | | | | | | |
| Month 0 to 3 | 2.0 (1.0) | 2.2 (1.7) | 2.3 (1.9) | 0.2 (−0.8, 1.1) | 0.67 | 0.3 (−0.6, 1.3) | 0.45 | 0.2 (−0.4, 0.8) | 0.60 |
| Month 4 to 6 | 1.6 (1.0) | 2.1 (1.7) | 2.0 (1.6) | 0.4 (−0.3, 1.1) | 0.26 | 0.3 (−0.4, 1.0) | 0.33 | −0.1 (−0.7, 0.5) | 0.78 |
| | n/N (%) | n/N (%) | n/N (%) | RD (95% CI) | p-Value | RD (95% CI) | p-Value | RD (95% CI) | p-Value |
| **Male sex partners without condoms in the past 3 months** | | | | | | | | |
| Month 0 to 3 | 2/7 (28.6) | 22/66 (33.3) | 26/68 (38.2) | 4.8 (−37.1, 32.4) | 0.98 | 9.7 (−32.8, 37.4) | 0.81 | 4.9 (−11.7, 21.2) | 0.60 |
| Month 4 to 6 | 5/13 (38.5) | 20/45 (44.4) | 24/62 (38.7) | 6.0 (−26.2, 34.1) | 0.82 | 0.3 (−31.5, 26.8) | 1.00 | −5.7 (−24.8, 13.5) | 0.61 |
| **Anal group sex in the past 3 months** | | | | | | | | |
| Month 0 to 3 | 2/7 (28.6) | 5/74 (6.8) | 6/72 (8.3) | −21.8 (−63.1, 3.2) | 0.08 | −20.2 (−61.3, 5.1) | 0.10 | 1.6 (−7.9, 11.4) | 0.77 |
| Month 4 to 6 | 1/14 (7.1) | 5/51 (9.8) | 9/69 (13.0) | 2.7 (−24.9, 17.1) | 0.84 | 5.9 (−22.1, 19.1) | 0.70 | 3.2 (−9.8, 15.2) | 0.61 |
| **Ever used substances before or during sex in the past 3 months** | | | | | | | | |
| Month 0 to 3 | 3/7 (42.9) | 31/74 (41.9) | 40/72 (55.6) | −1.0 (−39.6, 32.7) | 0.99 | 12.7 (−26.3, 46.2) | 0.66 | 13.7 (−3.0, 29.7) | 0.11 |
| Month 4 to 6 | 5/14 (35.7) | 18/51 (35.3) | 33/69 (47.8) | −0.4 (−31.0, 25.9) | 1.00 | 12.1 (−19.0, 37.4) | 0.47 | 12.5 (−5.8, 29.9) | 0.17 |
| **Tested for HIV** | | | | | | | | |
| Month 0 to 3 | 7/7 (100.0) | 60/74 (81.8) | 55/72 (76.4) | −18.9 (−31.3, 25.5) | 0.26 | −23.6 (−36.6, 20.4) | 0.17 | −4.7 (−18.3, 8.8) | 0.53 |
| Month 4 to 6 | 14/14 (100.0) | 48/51 (94.1) | 62/69 (89.9) | −5.9 (−17.1, 18.6) | 0.46 | −10.1 (−20.6, 15.8) | 0.24 | −4.3 (−14.9, 7.5) | 0.56 |
| Overall | 19/20 (95.0) | 83/90 (92.2) | 80/90 (88.9) | −2.8 (−12.4, 18.3) | 0.82 | −6.1 (−16.5, 14.7) | 0.50 | −3.3 (−12.9, 5.7) | 0.53 |
| **Tested for other STIs[b]** | | | | | | | | |
| Month 0 to 3 | 1/7 (14.3) | 6/74 (8.1) | 6/72 (8.3) | −6.2 (−49.1, 11.2) | 0.79 | −6.0 (−49.2, 11.5) | 0.86 | 0.2 (−9.7, 10.1) | 1.00 |
| Month 4 to 6 | 4/14 (28.6) | 3/51 (5.9) | 9/69 (13.0) | −22.7 (−51.7, −0.6) | 0.02 | −15.5 (−44.8, 6.6) | 0.15 | 7.2 (−4.7, 18.4) | 0.22 |
| Overall | 8/20 (40.0) | 14/90 (15.6) | 20/90 (22.2) | −24.4 (−48.5, −0.9) | 0.02 | −17.8 (−42.5, 5.5) | 0.10 | 6.7 (−5.2, 18.4) | 0.28 |

MD = mean difference. RD = risk (probability) difference expressed as a percentage.

[a]Due to rounding error, mean differences may differ by one decimal place from differences obtained by subtracting means listed in table.

[b]Other STIs included gonorrhea, chlamydia, human papillomavirus, and herpes simplex virus.

CI, confidence interval; SOC, standard of care; SST, syphilis self-testing; STI, sexually transmitted infection.

Our study findings may be generalizable to the many other settings where MSM seek online public health information that can be accessed without providing identifying information [35,36]. Online platforms represent a major opportunity for reaching marginalized MSM and designing pragmatic interventions [37]. Online platforms can reach MSM in remote rural areas, MSM who do not tell doctors about their same-sex behaviors (an especially important consideration in low- and middle-income countries with severe stigma), and other vulnerable subsets of MSM. At the same time, the approach used in this study requires an online

connection, willingness to use an online platform for requesting testing, and a postal service to mail the kits. Our empirical generalizability research found that MSM population characteristics from an online MSM trial were comparable to data from a national, cross-sectional sample of MSM [38].

We used rapid dual self-testing for syphilis and HIV in this study, given the syndemic of syphilis and HIV among MSM globally [38]. HIV self-testing has already created extensive infrastructure and public health pathways, including online testing programs, verification methods, integration of public health and community-based organization programs, and communication materials [20,21]. Previous studies have shown the feasibility and necessity of integrating SST into HIV self-testing services [4,21]. WHO also recommended using dual rapid tests for HIV and syphilis as the initial screening test in antenatal care [39]. Our study suggests that integrating HIV and SST could contribute to public health interventions focused on HIV and syphilis among MSM [40].

This study has implications for research and implementation. First, syphilis is a major public health problem, but is often overlooked and underfunded, especially in low- and middle-income countries. Our study expands the evidence base for self-care interventions and would provide valuable data for self-care programs and research. Increasing syphilis and HIV self-testing through self-care interventions may help address the syndemic of syphilis and HIV among MSM globally [38]. These self-testing programs could also be further expanded to include self-care interventions for other STIs (e.g., chlamydia self-collection and gonorrhea self-collection) [41]. Second, SST could expand syphilis testing among MSM. The ongoing COVID-19 pandemic restrictions on facility-based testing highlight the importance and potential of promoting SST. Third, from a financial perspective, SST can reduce costs compared with facility-based testing, enabling savings in syphilis testing programs to be reinvested in delivery of syphilis or other STI related services. This cost reduction is particularly relevant to many low- and middle-income countries with limited funding for non-HIV STI prevention services. Fourth, research on linkage to clinical services is essential to ensure the full benefit of self-testing approaches.

Several important limitations merit discussion. First, the impact of COVID-19 restrictions during the study period may have accentuated the demand for decentralized testing, as access to facility-based testing likely decreased. An online survey conducted among MSM across 31 provinces in China estimated that there was a 59% (95% CI: 58 to 60%) decrease in the number of MSM undergoing facility-based HIV testing in the first quarter of 2020 compared to the first quarter of 2019 [41]. A sensitivity analysis conducted by the coauthor AMW on the uptake of syphilis testing in the standard of care arm (see Methods) revealed no evidence of a difference (RD: −3.6%; 95% CI: −12.6% to 5.4%) between the original and the imputed dataset involving a 22% increase in testing uptake (S4 Table). Given the ongoing impacts of the COVID-19 pandemic on health facilities and preferences for self-testing, our study demonstrates how self-testing can meet the demand for syphilis testing among MSM through decentralized testing models. Second, while we present the first economic evaluation of SST using data from this RCT, future studies to evaluate the costs of scaling up this model and the long-term impact on syphilis epidemiology is warranted to confirm the value of SST. Third, although we used an objective measure for our primary outcome (photo verification), there is potential to underestimate the number of people testing for syphilis if they did not upload their syphilis test result. Fourth, we have data from treponemal test results because these test kits are commercially available, accurate, and field tested. The addition of nontreponemal tests [42] will be important for differentiating previous and new infections. Fifth, we used dual self-testing for syphilis and HIV in this study, which might raise the concern that the increase in the proportion of testing uptake is due to participants interested in the HIV testing. In this

study, we identified 18 men living with HIV, and 8 men were newly identified through participation in this study. We conducted a sensitivity analysis by removing the 18 men from the study and found little change to the original outcomes (results not shown). Sixth, there might be a risk of inadvertent disclosure of requesting a syphilis test.

In conclusion, this RCT demonstrates the effectiveness of SST to substantially increase syphilis test uptake among MSM in China. Further evaluation of SST is warranted to confirm its role in controlling syphilis in other contexts.

## Supporting information

**S1 CONSORT Checklist. CONSORT 2010 checklist of information to include when reporting a randomized trial.** * **CONSORT, Consolidated Standards of Reporting Trials.** (DOCX)

**S1 CONSERVE Checklists. Use CONSERVE–CONSORT for completed trial reports and CONSERVE-SPIRIT for trial protocols.** CONSORT, Consolidated Standards of Reporting Trials.
(DOCX)

**S1 Study Protocol. Promoting routine syphilis screening among MSM in China: study protocol for a RCT of SST and lottery incentive.** MSM, men who have sex with men; RCT, randomized controlled trial; SST, syphilis self-testing.
(DOCX)

**S1 Appendix. Complete economic evaluation. Table A:** Cost items included. **Table B:** Unit costs of SOC, SST, and lottery incentivized SST. **Fig A:** Decision tree model. **Fig B:** Tornado plot of the incremental cost per person tested for SST compared to SOC. **Fig C:** Cost-effectiveness acceptability curve for cost per person tested. **Fig D:** Tornado plot of incremental cost per person tested for lottery incentivized SST compared to SST. **Fig E:** Tornado plot of incremental cost per person diagnosed for SST compared to SOC. **Fig F:** Tornado plot of incremental cost per person diagnosed for lottery incentivized SST compared to SST. **Fig G:** Cost-effectiveness acceptability curve for cost per person diagnosed. SOC, standard of care; SST, syphilis self-testing.
(DOC)

**S1 Text. Web link to an instructional video for the SST package.** SST, syphilis self-testing.
(DOCX)

**S1 Fig. Manufacturer-supplied step-by-step instructions for the SST package.** SST, syphilis self-testing.
(TIF)

**S2 Fig. Result report card for the standard SST arm. SST, syphilis self-testing.**
(TIF)

**S3 Fig. Result report card for the lottery incentivized self-testing arm.**
(TIF)

**S1 Table. Participants' geographic distribution.**
(DOCX)

**S2 Table. Baseline characteristics of study participants stratified by loss to follow-up in the SST RCT in China in 2020.** RCT, randomized controlled trial; SST, syphilis self-testing.
(DOCX)

**S3 Table. Sensitivity analysis on the uptake of syphilis testing among all participants.**
(DOCX)

**S4 Table. Newly identified syphilis infections and linkage to care among men who had syphilis testing during the trial.**
(DOCX)

**S5 Table. Syphilis testing by mode: self-test, facility-based test, or both.**
(DOCX)

## Author Contributions

**Conceptualization:** Cheng Wang, Jason J. Ong, Weiming Tang, Joseph D. Tucker.

**Data curation:** Cheng Wang, Peizhen Zhao, Ann Marie Weideman.

**Formal analysis:** Cheng Wang, Peizhen Zhao, Ann Marie Weideman, Fern Terris-Prestholt.

**Funding acquisition:** Joseph D. Tucker.

**Methodology:** Cheng Wang, Jason J. Ong, Weiming Tang, M. Kumi Smith, Michael Marks, Hongyun Fu, Weibin Cheng, Joseph D. Tucker.

**Project administration:** Joseph D. Tucker, Bin Yang.

**Software:** Ann Marie Weideman.

**Visualization:** Heping Zheng.

**Writing – original draft:** Cheng Wang, Jason J. Ong.

**Writing – review & editing:** Cheng Wang, Jason J. Ong, Ann Marie Weideman, Weiming Tang, M. Kumi Smith, Michael Marks, Hongyun Fu, Joseph D. Tucker.

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
