## [Editor Report · Decision Letter 0]

29 Jun 2021

Dear Dr Wang, 

Thank you for submitting your manuscript entitled "Expanding syphilis test uptake using rapid dual self-testing for syphilis and HIV among men who have sex with men: a multi-arm randomized controlled trial in China" for consideration by PLOS Medicine.

Your manuscript has now been evaluated by the PLOS Medicine editorial staff and I am writing to let you know that we would like to send your submission out for external peer review.

Please re-submit your manuscript within two working days, i.e. by Jul 01 2021 11:59PM.

Kind regards,

Beryne Odeny

Associate Editor

PLOS Medicine

---

## [Decision Letter · Decision Letter 1]

13 Oct 2021

Dear Dr. Wang,

Thank you very much for submitting your manuscript "Expanding syphilis test uptake using rapid dual self-testing  for syphilis and HIV  among men who have sex with men: a multi-arm randomized controlled trial  in China" (PMEDICINE-D-21-02784R1) for consideration at PLOS Medicine. 

[LINK]

In light of these reviews, I am afraid that we will not be able to accept the manuscript for publication in the journal in its current form, but we would like to consider a revised version that addresses the reviewers' and editors' comments. Obviously we cannot make any decision about publication until we have seen the revised manuscript and your response, and we plan to seek re-review by one or more of the reviewers. 

We expect to receive your revised manuscript by Oct 28 2021 11:59PM. Please email us (plosmedicine@plos.org) if you have any questions or concerns.

We look forward to receiving your revised manuscript. 

Sincerely,

Beryne Odeny, 

PLOS Medicine

plosmedicine.org

1) Abstract:

a) Please structure your abstract using the PLOS Medicine headings (Background, Methods and Findings, Conclusions).

i) Please combine Background and Objective into one section, “Background.”

ii) Please combine the Design, Setting, Participants, Measurements, Results and Limitations sections into one section, “Methods and findings”. Please ensure that all numbers presented in the abstract are present and identical to numbers presented in the main manuscript text.

b) Please report your abstract according to CONSORT for abstracts: http://www.consort-statement.org/extensions?ContentWidgetId=562

c) Please indicates the dates during which study enrollment and follow up occurred.

d) Please provide the number of participants in each group.

e) Please provide the number of participants lost to follow up in each group.

f) Please state that the analysis was intention to treat.

g) Please quantify the main results (with p values in addition to 95% CI).

h) Please include a summary of adverse events if these were assessed in the study.

2) Data availability statement: If the data are owned by a third party but freely available upon request, please note this and state the owner of the data set and contact information for data requests (web or email address). Note that a study author cannot be the contact person for the data.

3) Author summary - At this stage, we ask that you reformat your non-technical Author Summary. The Author Summary should immediately follow the Abstract in your revised manuscript. This text is subject to editorial change and should be distinct from the scientific abstract. The summary should be accessible to a wide audience that includes both scientists and non-scientists. Please see our author guidelines for more information: https://journals.plos.org/plosmedicine/s/revising-your-manuscript#loc-author-summary.

4) Thank you for completing the CONSORT checklist. Please align the reporting of your manuscript with the checklist and replace the page numbers with paragraph numbers per section (e.g. "Methods, paragraph 1"), since the page numbers of the final published paper may be different from the page numbers in the current manuscript.

5) Please also report your study in line with the CONSERVE 2021 Statement checklist, a guideline for reporting trials which have been disrupted by COVID-19 and other circumstances. The guideline can be found at https://pubmed.ncbi.nlm.nih.gov/34152382/ . In the methods, please indicate the study has been reported according to the CONSERVE 2021 statement, and provide the checklist as a supplementary file.

6) Please specify whether informed consent was obtained and whether it was written or oral.

7) Please remove the “Role of the funding source”, “Declaration of interests”, and “Data sharing” from the main text. This information is captured in the metadata obtained in the submission form.

8) In the Methods and Results section:

a) Please provide 95% CIs and p values for estimates in the main text and tables

b) When a p value is given, please specify the statistical test used to determine it.

9) Figures and tables:

a) Please rename Figure 2, CONSORT flow chart

b) Please indicate in the figure caption the meaning of the whiskers in Figures

c) Please provide definitions for the following abbreviations: IC, AEs, EU/mL

d) Please ensure that Figure 1 complies with our figure requirement: http://journals.plos.org/plosmedicine/s/figures. 

e) Please confirm that the appropriate usage rights apply to the use of this Figure 1. Please see our guidelines for map images: https://journals.plos.org/plosmedicine/s/figures#loc-maps

10) References:

a) Please select the PLOS Medicine reference style in your citation manager. In-text reference call outs should be presented as follows noting the absence of spaces within the square brackets: "... countries [1,2]."

b) In the reference section, please ensure six names appear before et al.

c) Please ensure that journal name abbreviations consistently match those found in the National Center for Biotechnology Information (NCBI) databases. https://journals.plos.org/plosmedicine/s/submission-guidelines#loc-references. 

d) Please include access dates for all weblinks (e.g., Ref #32, and ensure that all weblinks are current and accessible.

11) To help us extend the reach of your research, please provide any Twitter handle(s) that would be appropriate to tag, including your own, your coauthors’, your institution, funder, or lab.

Comments from the Academic Editor:

Please pay special attention to reviewer concerns regarding: 1) the “dual intervention and 2) drop-out rates and availability of generic information on those who did not proceed with the rest of the study.

The authors could address the first point (about dual HIV/syphilis testing) as a limitation to the study and just comment on its possible implications. The second point (very high rate of self-exclusion) is more problematic, as it does call into question the generalizability of the results. This needs to be addressed in the analysis.

Three other comments from the Academic editor:

1. There is no discussion at all of the confidentiality/unintentional disclosure risk of requesting a syphilis test. I assume that there must be some stigma associated with STIs. It would improve the paper to learn something about this.

2. Related to point 1, there is no mention of the yield (% positive) for any of the arms. It's reasonable to guess that those actively seeking a test are at higher risk than those not. This could differ by arm; in any case it almost surely differs among those volunteering to participate and those not. The yield also affects the cost-effectiveness. This needs to be discussed, if not incorporated into the analysis.

3. The cost analysis did not capture the "full economic cost" as stated. It captured the financial cost to the provider. This is fine in terms of methods but a full economic evaluation should not be claimed. Including the cost of developing the lottery is odd, as this appears to be a research-specific cost that would not be incurred if the intervention were to be adopted for routine use. I would take that out.

Comments from the reviewers:

Reviewer #1: Thanks for the opportunity to review your manuscript. My role is a statistical reviewer so my comments focus on the study design, analysis and data presented. I have put general comments first, and followed these with queries specific to a section of the manuscript.

The study is a 3-arm parallel RCT that compares three strategies (usual care, self-testing, and self-testing with a 'lottery' incentive) in improving rates of syphilis testing in MSM in China. The study showed much higher rates of testing where self-resting was offered, and higher rates again with incentives at the final follow-up.

Was a statistical analysis plan prepared for the trial before data were unblinded to the investigators? What led to the changes in the statistical analysis specified in the protocol (a GEE model)? 

In the protocol, it was specified that if outcome measures were missing for 11-<20% of the participants, a sensitivity analysis with multiple imputation would be used. It looks based on Figure 2 that this is the case for the 6 months follow-up outcome. What was the results of this? For 3 months follow-up it is very close to 11% missing data - it would strengthen the reporting of the study if an appropriate MI procedure was completed with 3-month follow-up data as well.

One thing to clarify - how would patients in standard care arm be able to provide a photo verification of testing? Are patients routinely given results via email/message or are they given at in-person appointments? Could this change the ascertainment of outcomes in this usual care group? 

Was there any generic information captured before consent about who did not proceed with the rest of the study? The drop-out rate at this point is fairly high and I was interested in the reasons why so many people dropped out at this point.

P2, Abstract, Results. It's unclear at first glance what the '48.7%' refers to - maybe rephrasing as 'absolute difference in proportions' would be clearer? In tables, it's referred to as risk difference which would also work.

P5, Recruitment. Just to clarify - this was advertised on Blued, but users could share the link via other social media?

 P9, Sample Size. Setting the number of participants based on >99% looks a bit odd - was the sample size for the study fixed and then the power calculated after this had been decided? I don't have any objections to the sample size here, 120 per arm is not an outrageous number and the study arms are unlikely to have any detrimental effect on participants, it's just unusual to plan a sample size with power >0.9. The sample size information presented here is different to that in the protocol - what was the reason for the update? 

P9, General Analysis. 'modified intention to treat' is a variable term that I think is almost meaningless. It would be better to be straightforward and say exactly what the assumptions were, e.g. analysed as randomised to the extent we had at least one available visit, with list-wise deletion

P10, Analysis of Primary Outcome. Usually I would say Clopper-Pearson has undesirable qualities (i.e. conservative relative to more modern estimators) but with the sample size, and high prevalence of the outcomes it is safely applied here.

P11. What about the comparison of SST vs SST + incentives at 3 months? 

Table S2. This is a useful table - clearly the participants lost to follow-up were different on some of the key variables. When looking at the sensitivity analysis with imputation there are some good auxiliary variables that could be included based on this table. 

Table S4. This wasn't specified in the protocol - when was this sensitivity test planned? This type of imputation (similar to NRI) is an ok sensitivity test but the sensitivity test in the protocol (with MI) is more likely to give realistic results (assuming MAR).

Reviewer #2: The manuscript by Wang et al assesses the acceptability of self-testing for syphilis and HIV using a rapid test in a multi-arm randomized trial involving men who have sex with men in China. When compared to standard laboratory-based testing of specimens collected during a health visit, the option self-testing at home increased the proportion of participants tested from 14.7% to over 63%.

Specific comment

Figure 2. The box listing those exclusion states that '55 were currently living with syphilis.' This gives the impression that syphilis is like HIV regarding the lack of curable treatments. Consider using the same statement in the text (page 11 stating that 55 currently had syphilis).

Reviewer #3: This RCT study aimed to examine whether providing syphilis self-testing kits would increase the proportion of syphilis uptake among MSM in China. The results of this study are significant and had good implications for public health improvement. This manuscript is well written. 

However, the main concern is that the self-testing kits is dual syphilis/HIV self-testing kits, thus we do not know whether the increase of the proportion of the kits uptake is due to participants'interest in HIV testing or syphilis testing. The intervention is kind of a "dual intervention", while the authors only focused on part of the intervention and its result, which is not appropriate. 

Here are other comments need to be addressed:

1.In China, actually there is no free HIV and syphilis self-testing kits available. Thus, the SST arm provided free kits is a big offer for MSM, and could be recommended for public health policy makers, but why the authors design an SST +incentive arms? Since it needs more budget for government, and the implication is kind of far from the reality. 

2.The conclusion stated that :"…..particularly among men who had never tested for syphilis. I am thinking whether this conclusion is appropriate or not, since the enrollment is not stratified by non-testers or former testers. 

3.The big difference between the two intervention groups with the standard of care group partly due to the social isolation during the COVID-19 outbreak during Jan to May 2020. Most of the places strongly recommended people to stay home, and some of the VCTs are closed, for example, those run by gay-friendly CBOs. Thus the last sentence on page 4 stated that facility-based testing is still available is not always the truth, since the participants were from 124 cities, and the lockdown measures might not be the same. 

4.Another concern is that, 54082 participants clicked the flyer link, and 52089 withdrew before consent, which means that most of the participants were not interested in this program. Thus the external validity of the study intervention need to be considered. 

5.Among the 2124 men who are interested in this study, 1362 reported no condomless anal sex during the previous year, which is much different from the general MSM population. 

6. On page 5, the cost of the kit has a range, which means on market the kit has range of cost, or in this study? Typically the cost of the kit in this study should be same. 

7.What do you mean by "online informed consent"? do they provided written informed consent? 

8.How many of kits has been delivered to the participants per month (and in total)? And how much proportion of those kits were tested for themselves, and how much for partners or others?

[LINK]

---

## [Decision Letter · Decision Letter 2]

22 Dec 2021

Dear Dr. Wang,

Thank you very much for re-submitting your manuscript "Expanding syphilis test uptake using rapid dual self-testing  for syphilis and HIV  among men who have sex with men: a multi-arm randomized controlled trial  in China" (PMEDICINE-D-21-02784R2) for review by PLOS Medicine.

I have discussed the paper with my colleagues and the academic editor and it was also seen again by two reviewers. I am pleased to say that provided the remaining editorial and production issues are dealt with we are planning to accept the paper for publication in the journal.

[LINK]

We look forward to receiving the revised manuscript by Dec 22 2021 11:59PM.   

Sincerely,

Beryne Odeny, 

PLOS Medicine

plosmedicine.org

Requests from Editors:

1) With regard to data access, do you have contact information (web or email address) for a local authority in China that can grant permission?

2) Please add line numbers in the next revision

3) Author summary 

a) Under “Why was this study done?”, 2nd line should read “….expand syphilis testing”

b) adapting the author summary to present the findings in a narrative way rather than repeating all the numbers already in the abstract. 

4) In the abstract and main text, please rephrase “win $15 if they tested for syphilis” to “…win $15 if they had a syphilis test or similar…”

5) Please add p values (including “p<0.001”) after the CIs

6) Please do not report P<0.0001; report as P < 0.001. 

Comments from the Academic Editor:

I continue to have doubts about generalizability. I would like to see a paragraph in the discussion that addresses the generalizability of results to the entire potentially eligible population and makes clear that the sampling strategy, very large number of self-excluders, potential for disclosure, etc. all limit the applicability of results to a subset of the MSM population who are willing to be identified as wanting a syphilis/HIV test. As it stands, there is an implication that the findings would be relevant for the entire population, which is not correct.

Comments from Reviewers:

Reviewer #1: Thanks for the revised manuscript and response to my original queries. Overall, the updates address almost of my original queries (see below for a change with the missing data sensitivity analyses).

The extra information about recruitment (from the splash screen of the app) explains the drop-out rate. From my involvement with other work with similar approaches to recruitment via an app (although with a different target population) there was low 'click-through' rate to consent to recruitment. There probably isn't an easy way around this, the views of the subject-matter reviewers familiar with the population are important here in establishing if generalisability is likely to be an issue.

The sensitivity analyses around the missing data are reassuring. For the MI sensitivity analysis some additional detail is needed in the methods (e.g. number of imputations, rationale for the number of imputations chose, what variables were used in the imputation model). Table S4 (or another table) should include a repeat of the main analysis with the imputed datasets. Given the summary data it's very unlikely the conclusions would be different from the main analysis but it should still be included.

Reviewer #3: 

Most of the comments have been addressed. However, there are a couple concerns for authors' consideration.

1. In the abstract: The conclusion of "Compared to standard-of-care, providing SST significantly increased the proportion of MSM testing for syphilis in China and was cheaper (per person tested)."---- Since the "SST is short for "syphilis self testing", then I think "providing SST" should be specified as "providing SST plus HIV self-testing dual kits", since the study is not examining the effectiveness of providing single SST kits. 

And the same problem during the Author Summary and other parts of the main text. 

2. On page 16: "Previous studies have shown the feasibility and necessity of integrating syphilis selftesting into HIV self-testing services.4,24 WHO also recommended using dual rapid tests for HIV and syphilis as the initial screening test in antenatal care." --------With this information already in the literature, the authors should justify the significance of this study, since WHO has recommended and China also conducted in reality (many CDC and Community-based organizations used duo testing kits in daily practice).

3. What so called the "online informed consent" is more like an information notification rather than informed consent. Usually IRB requires written or oral IC. 

4. The inclusion criteria for having condomless anal sex is previous 12 months, but having no syphilis test is previous 6 months. Why the time point selection is different? What is the rational? [as stated by authors, in literature, usually they evaluated condomless sex in recent 6 months, 3 months and the last time]

[LINK]

---

## [Editor Report · Decision Letter 3]

20 Jan 2022

Dear Dr. Wang,

Thank you very much for re-submitting your manuscript "Expanding syphilis test uptake using rapid dual self-testing  for syphilis and HIV  among men who have sex with men: a multi-arm randomized controlled trial  in China" (PMEDICINE-D-21-02784R3) for review by PLOS Medicine.

I have discussed the paper with my colleagues and the academic editor. I am pleased to say that provided the remaining editorial and production issues are dealt with we are planning to accept the paper for publication in the journal.

[LINK]

We look forward to receiving the revised manuscript by Jan 27 2022 11:59PM.   

Sincerely,

Beryne Odeny, 

PLOS Medicine

plosmedicine.org

Requests from Editors:

1) Title: Please remove “in China” from the subtitle and place it right before the colon. It should read “Expanding syphilis test uptake using rapid dual self-testing for syphilis and HIV among men who have sex with men in China: a multi-arm randomized controlled trial”

2) Abstract: Please remove the following sentence from the end of the Methods and Findings section: “The trial is registered with the Chinese Clinical Trial Registry, number ChiCTR1900022409. The study design and results were reported according to CONSORT 2010 guidelines.”

3) Please replace “Chinese MSM” with "MSM in China."

4) Please ensure that journal name abbreviations consistently match those found in the National Center for Biotechnology Information (NCBI) databases. https://journals.plos.org/plosmedicine/s/submission-guidelines#loc-references. For example, PLOS Medicine should be PLOS Med.

[LINK]

---

## [Editor Report · Decision Letter 4]

25 Jan 2022

Dear Dr Wang, 

On behalf of my colleagues and the Academic Editor, Dr. Sydney Rosen, I am pleased to inform you that we have agreed to publish your manuscript "Expanding syphilis test uptake using rapid dual self-testing  for syphilis and HIV  among men who have sex with men in China: a multi-arm randomized controlled trial" (PMEDICINE-D-21-02784R4) in PLOS Medicine.

PRESS

Sincerely, 

Beryne Odeny  

PLOS Medicine